# Multifaceted Mechanisms of Action of Metformin Which Have Been Unraveled One after Another in the Long History

**DOI:** 10.3390/ijms22052596

**Published:** 2021-03-05

**Authors:** Hideaki Kaneto, Tomohiko Kimura, Atsushi Obata, Masashi Shimoda, Kohei Kaku

**Affiliations:** Department of Diabetes, Endocrinology and Metabolism, Kawasaki Medical School, 577 Matsushima, Kurashiki 701-0192, Japan; tomohiko@med.kawasaki-m.ac.jp (T.K.); obata-tky@med.kawasaki-m.ac.jp (A.O.); masashi-s@med.kawasaki-m.ac.jp (M.S.); kka@med.kawasaki-m.ac.jp (K.K.)

**Keywords:** metformin, AMPK, glucagon signaling, autophagy, GDF15, gut microbiome, mTOR, COVID-19

## Abstract

While there are various kinds of drugs for type 2 diabetes mellitus at present, in this review article, we focus on metformin which is an insulin sensitizer and is often used as a first-choice drug worldwide. Metformin mainly activates adenosine monophosphate-activated protein kinase (AMPK) in the liver which leads to suppression of fatty acid synthesis and gluconeogenesis. Metformin activates AMPK in skeletal muscle as well, which increases translocation of glucose transporter 4 to the cell membrane and thereby increases glucose uptake. Further, metformin suppresses glucagon signaling in the liver by suppressing adenylate cyclase which leads to suppression of gluconeogenesis. In addition, metformin reduces autophagy failure observed in pancreatic β-cells under diabetic conditions. Furthermore, it is known that metformin alters the gut microbiome and facilitates the transport of glucose from the circulation into excrement. It is also known that metformin reduces food intake and lowers body weight by increasing circulating levels of the peptide hormone growth/differentiation factor 15 (GDF15). Furthermore, much attention has been drawn to the fact that the frequency of various cancers is lower in subjects taking metformin. Metformin suppresses the mechanistic target of rapamycin (mTOR) by activating AMPK in pre-neoplastic cells, which leads to suppression of cell growth and an increase in apoptosis in pre-neoplastic cells. It has been shown recently that metformin consumption potentially influences the mortality in patients with type 2 diabetes mellitus and coronavirus infectious disease (COVID-19). Taken together, metformin is an old drug, but multifaceted mechanisms of action of metformin have been unraveled one after another in its long history.

## 1. Introduction

Pancreatic β-cell dysfunction and insulin resistance in insulin target tissues such as the liver, skeletal muscle and adipose tissues are the two main characteristics of type 2 diabetes mellitus. The number of subjects with type 2 diabetes mellitus is markedly increasing all over the world due to changes in lifestyle such as overeating and lack of exercise. Such an increase in subjects with type 2 diabetes has become a financial burden in many countries. So far, various kinds of drugs for type 2 diabetes mellitus have been developed, and at present, there are many kinds of anti-diabetic drugs from which we can choose depending on each patient’s pathophysiological conditions. Incretin-related drugs (dipeptidyl peptidase-IV (DPP-IV) inhibitors and glucagon-like peptide-1 receptor activators (GLP-1RA)) and sodium-glucose cotransporter 2 (SGLT2) inhibitors are relatively new drugs and have been drawing much attention in various aspects. In contrast, metformin is an old drug, but its pleiotropic mechanisms of action have been gradually clarified in its long history. There were times when the reputation of metformin was not very high, but due to various discoveries about new mechanisms of action of metformin, the Association for the Study of Diabetes (the American Diabetes Association and the European Association for the Study of Diabetes) consensus guideline on the management of type 2 diabetes stipulates that metformin should be used as a first-choice drug for type 2 diabetes mellitus. Indeed, it is very often used as a first-choice drug in clinical practice all over the world. In addition, since metformin is quite cheap compared to other anti-diabetic drugs, usage of metformin reduces the financial burden on subjects with type 2 diabetes mellitus. In this review article, we focus on metformin which is an old but marvelous drug.

## 2. Glucose Toxicity Is an Underlying Mechanism for Type 2 Diabetes Mellitus

It is well known that insulin is secreted from pancreatic β-cells and the main insulin target tissues are the liver, skeletal muscle and adipose tissues. Type 2 diabetes mellitus is characterized by β-cell dysfunction and insulin resistance. Chronic exposure of β-cells and insulin target tissues to hyperglycemia leads to the deterioration of β-cell function and aggravation of insulin resistance [1,2,3,4,5,6]. Such phenomena are well known as glucose toxicity. First, overeating and/or obesity lead to the development of insulin resistance. Although pancreatic β-cells produce and secrete insulin in response to high glucose concentrations under healthy conditions, β-cells are compelled to secrete larger amounts of insulin to compensate such increased insulin resistance under diabetic conditions. Once hyperglycemia becomes apparent and β-cells are chronically exposed to hyperglycemia, β-cell function is gradually deteriorated due to some grueling overwork. Insulin production and secretion are progressively reduced, accompanied by reduced expression of insulin gene transcription factors MafA [7,8,9,10,11,12] and PDX-1 [13,14,15,16,17,18]. Insulin signaling in insulin target tissues is also weakened by the burden of glucose toxicity, which leads to the aggravation of insulin resistance. Such debilitation of β-cell function and development of insulin resistance lead to further aggravation of type 2 diabetes mellitus. In clinical practice, it is very essential to alleviate such β-cell glucose toxicity in order to forestall the aggravation of diabetes mellitus.

In response to ingestion of food, glucagon-like peptide-1 (GLP-1) and glucose-dependent insulinotropic polypeptide (GIP) are released from the gastrointestinal tract, which augment glucose-stimulated insulin secretion, reduce β-cell apoptosis and facilitate β-cell proliferation. Under diabetic conditions, however, GLP-1 and GIP receptor expression levels are reduced, which is likely bothersome for β-cells [19,20,21]. While transcription factor 7-like 2 (TCF7L2) is an important transcription factor of GLP-1 and GIP receptors and plays a crucial role in the maintenance of β-cell function, decreased expression of TCF7L2 under diabetic conditions is likely involved in such reduction in both incretin receptors [22,23,24]. Even when GLP-1 and GIP are secreted and come close to β-cells, they cannot function fully due to a reduction in their receptor expression on the β-cell membrane. 

In addition, there are several reports showing the essence of insulin signaling in endothelial cells [25,26,27,28,29,30]. Insulin binds to insulin receptors in the endothelial cell membrane, which activates insulin signaling in endothelial cells. After the binding, insulin receptor substrate (IRS), phosphatidylinositol 3-kinase (PI3-K), 3-phosphoinositide-dependent protein kinase-1 (PDK1) and protein kinase B (Akt) are phosphorylated sequentially. Such activation of insulin signaling finally increases expression of endothelial nitric oxide synthase. Therefore, activation of insulin signaling in endothelial cells augments the amount of nitric oxide production, which finally leads to the increase in blood flow and angiogenesis in islets. Since endothelial cell dysfunction is observed under diabetic conditions, it is possible that endothelial dysfunction leads to hypoxia and ischemia through reduced production of nitric oxide. In addition, since pancreatic islets are particularly vulnerable to hypoxia and/or ischemia, endothelial dysfunction can more easily lead to aggravation of β-cell function compared to other cells or tissues. Indeed, we recently reported that in vascular endothelial-specific knockout mice of PDK1, one of the important molecules in insulin signaling, β-cell mass became smaller and β-cell function was impaired [30]. Such ablation of endothelial PDK1 reduced vascularity in islets, which led to hypoxia, ER stress and inflammation in β-cells. Therefore, we think that endothelial dysfunction is also, at least in part, involved in β-cell dysfunction found in type 2 diabetes mellitus.

## 3. Various Agents for Type 2 Diabetes Mellitus Protect Pancreatic β-Cells against Glucose Toxicity

There are various kinds of drugs for type 2 diabetes mellitus such as insulin secretagogues and insulin sensitizers. Among several insulin secretagogues, incretin-related drugs have most often been used in clinical practice. Dipeptidyl peptidase-IV (DPP-IV) inhibitors as well as glucagon-like peptide-1 receptor activators (GLP-1RA) enhance insulin secretion and suppress glucagon secretion, leading to amelioration of glycemic control. We have reported that DPP-IV inhibitors or GLP-1RA ameliorated glycemic control and protected β-cells against glucose toxicity in type 2 diabetic mice [31,32,33,34,35]. In addition, incretin-related drug liraglutide increased β-cell function and mass only at an early stage of diabetes but not at an advanced stage [35]. Only at an early stage, insulin biosynthesis and secretion were significantly enhanced by liraglutide, accompanied by augmentation of MafA and PDX-1 expression [35]. We think that it is very essential to use incretin-based drugs at an early stage of diabetes in order to make the most of such drugs. In addition, much attention has been drawn recently to the anti-arteriosclerosis effects of incretin-related agents in the basic research area [36,37,38,39,40,41] as well as in clinical practice [42,43,44,45,46,47,48,49,50,51,52]. In the basic research area, we recently reported that incretin expression was down-regulated under diabetic conditions [39] and that incretin-related drugs exerted more beneficial anti-arteriosclerosis at an early stage of diabetes [41]. There have been many large-scale clinical trials regarding the protective role of incretin-based agents against atherosclerosis or cardiovascular events in subjects with type 2 diabetes mellitus [42,43,44,45,46,47,48,49,50,51,52]. 

Metformin and thiazolidine are insulin sensitizers. Metformin, one of the insulin sensitizers, is often used as a first-choice drug worldwide. Metformin is known to have pleiotropic roles in a variety of tissues such as the liver and skeletal muscle, and a variety of mechanisms of its action have been elucidated so far, as described below in detail. Thiazolidine also has multifaceted effects such as enhancement of insulin sensitivity, β-cell protective effects, miniaturization of visceral fat cells, enhancement of adiponectin secretion and anti-arteriosclerosis effects [31,34,35,53,54]. 

Furthermore, recently, sodium-glucose cotransporter 2 (SGLT2) inhibitors have been drawing much attention in the diabetes research area as well as in clinical practice. SGLT2 inhibitors function in an insulin-independent manner and ameliorate glycemic control through an increase in urinary glucose excretion. We have reported that SGLT2 inhibitors protect β-cells against glucose toxicity in type 2 diabetic mice [55,56,57]. Indeed, SGLT2 inhibitors increased insulin biosynthesis and glucose-stimulated insulin secretion, as well as increasing β-cell mass through the reduction in β-cell apoptosis and the enhancement of β-cell proliferation. In addition, we recently showed that SGLT2 inhibitor luseogliflozin exerted more protective effects at an early stage of diabetes compared to an advanced stage. Furthermore, we reported that longer-term use of luseogliflozin exerted more beneficial effects on pancreatic β-cell function and mass compared to short-term use [57]. In addition, since several potential side effects of SGLT2 inhibitors, about which many clinicians were previously concerned, have substantially been wiped out at present, we should start SGLT2 inhibitors at an early stage of diabetes in subjects to whom therapy with SGLT2 inhibitors is thought to be appropriate in clinical practice as well. SGLT2 inhibitors are known to exert beneficial effects on insulin target tissues such as the liver, skeletal muscle and adipose tissues. Indeed, it was reported that SGLT2 inhibitors improved muscle insulin sensitivity, although it enhanced endogenous glucose production, and that SGLT2 inhibitors improved insulin resistance in skeletal muscle and accelerated lipolysis in adipose tissues [58,59,60,61]. Furthermore, it has been elucidated that SGLT2 inhibitors have preventive effects on heart failure and proteinuria and thereby have cardio-protective and renal protective effects, both of which have drawn much attention recently, although we did not describe these points in detail in this review article. 

## 4. Metformin Activates Adenosine Monophosphate-Activated Protein Kinase (AMPK) in the Liver and Skeletal Muscle Which Leads to Suppression of Gluconeogenesis in the Liver and Increase in Glucose Uptake into Skeletal Muscle

Metformin enhances insulin sensitivity and ameliorates glycemic control mainly through a reduction in hepatic glucose production and enhancement of glucose utilization. AMPK is one of the major cellular regulators for glucose and lipid metabolism. It was reported that metformin activated AMPK in the liver, leading to a reduction in acetyl-CoA carboxylase (ACC), enhancement of fatty acid oxidation and suppression of lipogenic enzyme expression [62,63,64,65]. Metformin-mediated AMPK activation suppresses expression of sterol regulatory element binding protein-1c (SREBP-1), an important lipogenic transcription factor, leading to suppression of fatty acid synthesis (Figure 1). Further, while phosphoenolpyruvate carboxykinase (PEPCK) and glucose 6-phosphatase (G6Pase) are key gluconeogenic enzymes, metformin-mediated AMPK activation reduces both enzymes’ expression, leading to suppression of gluconeogenesis in the liver. Metformin also activates AMPK in skeletal muscle which increases translocation of glucose transporter 4 to the cell membrane and thereby increases glucose uptake. These effects finally ameliorate fatty liver and insulin resistance. It was reported recently that metformin inhibited mitochondrial respiratory complex I, leading to an increase in the ratio of adenosine monophosphate (AMP) to adenosine triphosphate (ATP). Such alteration likely leads to inactivation of AMPK [63]. It was also reported that metformin inactivated mitochondrial glycerol-3-phosphate dehydrogenase which was likely involved in suppression of gluconeogenesis in the liver [64].

## 5. Metformin Suppresses Glucagon Signaling in the Liver by Suppressing Adenylate Cyclase Which Leads to Suppression of Gluconeogenesis in the Liver

Glucagon is secreted from pancreatic α-cells and functions as one of the counter-regulatory hormones, leading to an increase in blood glucose levels. As one main mechanism of glucagon action, it is known that glucagon binds to the glucagon receptor in the liver, which activates adenylate cyclase and converts adenosine triphosphate (ATP) to cyclic AMP (cAMP). Increased cAMP activates protein kinase A (PKA), which facilitates gluconeogenesis. Thereby, glucagon leads to the aggravation of glycemic control. It was reported that metformin antagonized such action of glucagon, which led to amelioration of glycemic control. Metformin treatment led to the accumulation of AMP, which finally inhibited adenylate cyclase. Inactivation of adenylate cyclase reduced cyclic AMP levels and PKA activity and suppressed glucagon signaling, leading to suppression of gluconeogenesis (Figure 2) [66]. These findings clearly support the new mechanism of action of metformin as a suppressor of glucagon signaling in the liver. In addition, it was reported recently that metformin inhibited mitochondrial respiratory complex I, leading to an increase in the AMP/ATP ratio. Such alteration likely inactivates adenylate cyclase activity, leading to suppression of glucagon signaling in the liver [63]. Thereby, such inhibition of mitochondrial respiratory complex I suppresses gluconeogenesis through activation of AMPK, as well as suppressing glucagon signaling through inactivation of adenylate cyclase activity. Such alteration leads to amelioration of glucose metabolism and a reduction in insulin resistance in the liver, which finally leads to amelioration of glycemic control. 

## 6. Metformin Reduces Autophagy Failure Observed in Pancreatic β-Cells under Diabetic Conditions

Autophagy is involved in a variety of phenomena in our body and has been paid much attention to in various research areas. For example, in the diabetes research area, it has been reported, so far, that autophagy failure is observed in pancreatic β-cells under diabetic conditions [67,68,69,70,71,72]. In the process of autophagy, formation of autophagosomes and proteolysis of autolysosomes are main and important steps. When the autophagy system functions normally, insulin resistance enhances autophagy in β-cells, which finally leads to compensatory hypertrophy of β-cells. However, when the autophagy system does not function well, autophagy in β-cells is not enhanced by insulin resistance and compensatory hypertrophy of β-cells is not observed. Further, in human pancreatic islets with type 2 diabetes mellitus, larger numbers of vacuoles were observed compared to the control, suggesting an increase in autophagosomes. In addition, expression levels of various lysosome-related enzymes were reduced under diabetic conditions. These data suggest that autophagy failure is involved in β-cell dysfunction found in type 2 diabetes mellitus.

Furthermore, it was reported that metformin reduced autophagy failure observed in pancreatic β-cells under diabetic conditions [72]. When healthy β-cells were treated with free fatty acids (FFA), autophagosomes were increased and lysosome-related enzyme expression was reduced. However, when β-cells were treated with FFA and metformin, autophagosomes were not increased and lysosome-related enzyme expression was not reduced, indicating that the autophagy system was recovered. These data suggest that metformin mitigates pancreatic β-cell autophagy failure observed under diabetic conditions. 

## 7. Metformin Alters the Gut Microbiome and Glucose Absorption from the Intestine

Much attention has been drawn to the fact that alteration of the gut microbiome has a variety of influences on various tissues in our body. It was recently reported that metformin altered the gut microbiome which, at least in part, contributes to the therapeutic effects of metformin [73]. In a double-blind study, the authors randomized subjects with treatment-naive type 2 diabetes mellitus to metformin or placebo for 4 months and showed that metformin had strong effects on the gut microbiome. Furthermore, transfer of fecal samples from metformin-treated donors to germ-free mice showed that glucose tolerance was improved in mice that received metformin-altered microbiota. These findings support the idea that the altered gut microbiota is, at least in part, involved in the anti-diabetic effects of metformin. We think that the presence of such an underlying mechanism of metformin indicates that metformin would be very promising.

In addition, it was very recently reported that metformin altered glucose absorption from the intestine [74,75]. Indeed, positron emission tomography (PET)-computed tomography has shown that metformin facilitates the intestinal accumulation of [^18^F] fluorodeoxyglucose (FDG), a non-metabolizable glucose derivative. In this study, accumulation of [^18^F] FDG was evaluated in different portions of the intestine. As a result, [^18^F] FDG accumulation in the ileum and hemicolon was also larger in subjects with metformin. Furthermore, the maximum standardized uptake value for the intraluminal space of the ileum and hemicolon was larger in subjects with metformin. Taken together, metformin treatment is likely associated with increased accumulation of [^18^F] FDG in the intraluminal space of the intestine, indicating that metformin facilitates the transport of glucose from the circulation into excrement.

## 8. Metformin Reduces Food Intake and Lowers Body Weight by Increasing Circulating Level of the Peptide Hormone Growth/Differentiation Factor 15 (GDF15)

Weight gain and obesity are serious global health concerns, and pharmacological therapies or bariatric surgery have been performed for subjects with severe obesity. Metformin is known to lower body weight, and thus it seems that metformin has a health benefit beyond its glucose-lowering effect. Although its molecular mechanism remained unclear for a long time, it has been reported recently that metformin increases circulating levels of GDF15, a member of the transforming growth factor β superfamily, which leads to a reduction in food intake and body weight [76,77,78,79,80,81,82,83,84,85]. GDF15 is produced by various cells responding to a variety of stresses or stimuli, and GDF15 functions through its receptor which is expressed in the hindbrain and thereby reduces food intake. A recent clinical study showed that there was a close association between metformin usage and circulating levels of GDF15. Recent basic research also demonstrated that metformin increased circulating GDF15 in mice, accompanied by an increase in GDF15 expression in the intestine, colon and kidney. In addition, metformin suppressed body weight gain in mice treated with a high-fat diet, but such phenomena were not observed in mice lacking GDF15. Similarly, such phenomena were not observed in mice lacking GDNF family receptor α-like (GFRAL), which is known as a receptor of GDF15. In mice treated with a high-fat diet, the weight reduction effects of metformin were reduced by a GFRAL-antagonist antibody, although the glucose-lowering effect of metformin was not influenced by that antibody. These data further strengthen the idea that GDF15 is involved in the reduction in food intake and body weight in subjects with type 2 diabetes mellitus who are treated with metformin.

## 9. Metformin Suppresses Mechanistic Target of Rapamycin (Mtor) by Activating AMPK in Pre-Neoplastic Cells and Thereby Suppresses the Onset and/or Development of Various Cancers

Much attention has been drawn recently to the fact that the frequency of various types of cancer in subjects with diabetes mellitus is higher compared to that in healthy subjects [86,87,88,89,90,91,92,93]. In particular, the frequency of hepatocellular carcinoma and colorectal cancer is higher under diabetic conditions compared to healthy conditions. Thus, malignancy has been recently regarded as one of diabetic complications in addition to acute and chronic complications such as microangiopathies (diabetic nephropathy, retinopathy and neuropathy) and macroangiopathies (ischemic heart diseases, stroke and arteriosclerosis obliterans). There are several possible reasons why the frequency of malignancy is increased under diabetic conditions. First, chronic hyperglycemia increases various inflammatory cytokines and thereby activates nuclear factor-kappa B (NF-kB) and/or signal transducer and activator of transcription 3 (STAT3), which finally leads to the onset of neoplastic cells. Second, hyperinsulinemia, which is often observed in obese subjects with type 2 diabetes mellitus, activates insulin receptors in pre-neoplastic cells, leading to the onset of neoplastic cells. In addition, hyperinsulinemia decreases expression of insulin-like growth factor binding proteins 1 and 2 (IGFBP1 and IGFBP2) and thereby activates insulin-like growth factor-1 (IGF-1), which could also lead to the onset of neoplastic cells.

Furthermore, attention has been drawn to the fact that the frequency of various cancers is lower in subjects taking metformin. Indeed, there is a large amount of clinical evidence showing the possibility that usage of metformin decreases the risk of neoplastic transformation and enhances the response to some chemotherapies [94,95,96,97,98,99,100]. Metformin suppresses mTOR by activating AMPK in pre-neoplastic cells which leads to suppression of cell growth and an increase in apoptosis in pre-neoplastic cells (Figure 3) [94,95]. It seems that metformin exerts potential anti-tumorigenic effects independently of its hypoglycemic effects. In general, insulin and IGF-1 activate PI3-K, Akt and mTOR, which finally leads to enhancement of cell growth and suppression of apoptotic cell death in pre-neoplastic cells. There are several potential mechanisms concerning how metformin can suppress the development of neoplastic cells. First, metformin activates the AMPK pathway in pre-neoplastic cells which leads to suppression of mTOR activation. Such a pathway finally leads to suppression of cell growth and an increase in apoptosis in pre-neoplastic cells. Second, since metformin is an insulin sensitizer, it reduces circulating insulin levels, which is also, at least in part, involved in the anti-tumorigenic effects of metformin. Inhibition of protein synthesis, inhibition of the unfolded protein response (UPR), activation of the immune system and eradication of cancer stem cells are also possibly involved in the anti-tumorigenic effects of metformin. 

Taken together, while the frequency of various types of malignancies in subjects with diabetes mellitus is higher compared to that in healthy subjects, much attention has been drawn to the fact that the frequency of various cancers is lower in subjects taking metformin.

## 10. Metformin Consumption Potentially Influences the Mortality in Subjects with Type 2 Diabetes Mellitus and Coronavirus Infectious Disease (COVID-19)

Coronavirus infectious disease (COVID-19) has caused a new pandemic all over the world. The mortality in patients with COVID-19 is extremely high, and the main reason for deaths is severe pneumonia [101]. In subjects with COVID-19, large amounts of inflammatory cytokines are produced which likely causes a cytokine storm and is involved in the development of various complications such as serious pneumonia. The defense mechanism and immune system against inflammation are very vulnerable in senior subjects or subjects with diabetes mellitus, respiratory tract diseases, malignancy or coronary heart disease. Therefore, the infection risk and severity become very high in such subjects with comorbidities. It was reported that the mortality was very high in subjects with both COVID-19 and diabetes mellitus [102,103]. Since it is known that diabetic subjects have low-grade inflammation, we assume that such inflammation is, at least in part, involved in the vulnerability of diabetic subjects to COVID-19 and the severity of COVID-19 under diabetic conditions. 

It has been reported that metformin is associated with lower mortality in subjects with COVID-19 and diabetes mellitus [104,105]. In that study, several medical databases (Pubmed, EuropePMC, EBSCOhost, Proquest, Cochrane library) and two health science preprint servers (preprint.org and Medrxiv) were systematically searched for relevant literature. As a result, the meta-analysis with more than 10,000 subjects showed that metformin was associated with lower mortality in a pooled non-adjusted model (odds ratio (OR), 0.45; confidential interval (CI), 0.25–0.81) and a pooled adjusted model (OR, 0.64; CI, 0.43- 0.97). The analysis clearly indicates that metformin consumption is closely associated with lower mortality in subjects with COVID-19.

There are several possible mechanisms concerning how metformin exerts beneficial effects on mortality in subjects with COVID-19. First, it is known that metformin reduces pro-inflammatory cytokine levels such as tumor necrosis factor-α or interleukin-6. In addition, it was shown that metformin had some beneficial effects on viral infections such as hepatitis C virus or severe acute respiratory syndrome coronavirus 2 [106,107,108]. Therefore, it is possible that metformin has some favorable effects on COVID-19 by altering inflammatory cytokine levels. Second, as described above, metformin alters the gut microbiome and mitigates autophagy failure, both of which likely lead to the activation of the immune response and defense mechanism against an inflammatory cytokine storm. Finally, as described above as well, it is known that metformin blocks the mTOR pathway, while mTOR plays a crucial part in the pathogenesis of influenza and Middle East respiratory syndrome coronavirus infection. Therefore, it is possible that blocking the mTOR pathway by metformin, at least in part, contributes to the beneficial effect of metformin on the mortality in subjects with type 2 diabetes mellitus and COVID-19. Although further studies are necessary to conclude such possible mechanisms of action of metformin, the possibility that metformin is associated with lower mortality in subjects with COVID-19 and diabetes mellitus brings some emerging hope to us amid the current worldwide pandemic situations caused by COVID-19. 

## 11. Conclusions

In this review article, we featured various mechanisms of action of metformin which have been elucidated so far. First, metformin activates AMPK in the liver which leads to suppression of fatty acid synthesis and gluconeogenesis. Metformin also activates AMPK in skeletal muscle which increases translocation of glucose transporter 4 to the cell membrane and thereby increases glucose uptake. Second, metformin suppresses glucagon signaling in the liver by suppressing adenylate cyclase which leads to suppression of gluconeogenesis. Third, metformin reduces autophagy failure observed in pancreatic β-cells under diabetic conditions. Fourth, metformin alters the gut microbiome and glucose absorption from the intestine and facilitates the transport of glucose from the circulation into excrement. Fifth, metformin reduces food intake and lowers body weight by increasing circulating levels of GDF15. Sixth, much attention has been drawn to the fact that the frequency of various cancers is lower in subjects taking metformin. Metformin suppresses mTOR by activating AMPK in pre-neoplastic cells, which leads to suppression of cell growth and an increase in apoptosis in pre-neoplastic cells. Finally, while COVID-19 has caused a new pandemic all over the world, it has been reported recently that metformin consumption potentially influences the mortality in subjects with COVID-19 and type 2 diabetes mellitus, which brings great hope to us amid the current worldwide pandemic caused by COVID-19. Taken together, metformin is a medicine with a long history, but the multifaceted mechanisms of action of metformin have been elucidated one after another in its long history, and the usefulness of metformin is very promising from clinical points of view as well as in the basic research area. 

## Figures and Tables

**Figure 1 ijms-22-02596-f001:**
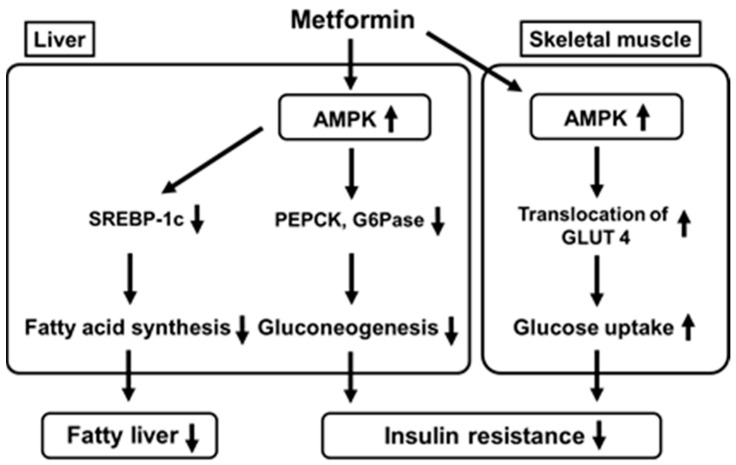
Metformin activates AMP-activated protein kinase (AMPK) in the liver which leads to suppression of fatty acid synthesis and gluconeogenesis. Metformin also activates AMPK in skeletal muscle which increases translocation of glucose transporter 4 to the cell membrane and thereby increases glucose uptake. SREBP-1c, sterol regulatory element binding protein-1c; PEPCK, phosphoenolpyruvate carboxykinase; GAPase, glucose 6-phosphatase; GLUT 4, glucose transporter 4.

**Figure 2 ijms-22-02596-f002:**
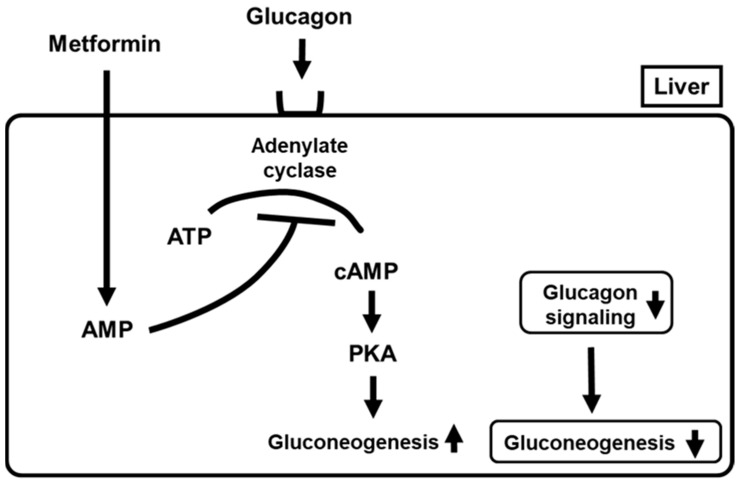
Metformin suppresses glucagon signaling in the liver by suppressing adenylate cyclase which leads to suppression of gluconeogenesis. PKA, protein kinase A.

**Figure 3 ijms-22-02596-f003:**
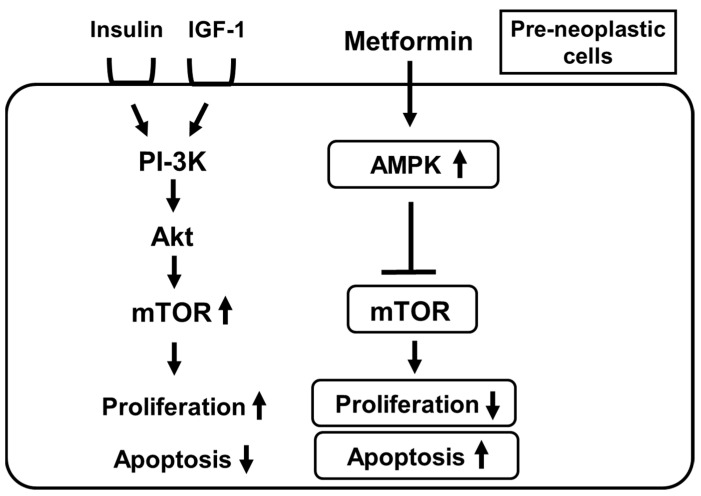
Metformin suppresses the mechanistic target of rapamycin (mTOR) by activating AMPK in pre-neoplastic cells which leads to suppression of cell growth and an increase in apoptosis in pre-neoplastic cells. IGF-1, insulin-like growth factor; PI-3K, phosphatidylinositol-3 kinase; Akt, protein kinase B.

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
