# Peer review of "Multifaceted Mechanisms of Action of Metformin Which Have Been Unraveled One after Another in the Long History"

_ijms, 2021, doi:10.3390/ijms22052596_

Round 1

Reviewer 1 Report

Dear Author

Author summarized the roles and mechanisms of Metformin in various cell types and organs.

Overall description for each section is well writen.

I only would like to recommend to put representative figure that show the summary for each section.

Thank you.

Jong-Seok Moon

Author Response

Thank you very much for your favorable comments.

We totally appreciate your comments.

The figures were missing by mistake in the previous version.

In the revise version, we did show Fig. 1, 2 and 3 in the manuscript. 

Reviewer 2 Report

The paper covers all the various mechanisms of action of Metformin, including immunological ones.

The article is consistent within itself. The references are relevant and recent. Appropriate and key studies are included. The paper is comprehensive, the flow is logical and the data is presented critically. The figures are clear, demonstrative, and beneficial.

There are no major drawbacks in the paper.

Author Response

Thank your very much for your favorable comments.

We totally appreciate your comments.

Round 2

Reviewer 1 Report

Dear Authors

Authors addressed all my requests.

I have no further question.

Thank you.